# Psychometric Properties of the Depression Anxiety Stress Scales (DASS-42 and DASS-21) in Patients with Hematologic Malignancies

**DOI:** 10.3390/jcm14062097

**Published:** 2025-03-19

**Authors:** Serkan Güven, Ertuğrul Şahin, Nursel Topkaya, Öznur Aydın, Sude Hatun Aktimur, Mehmet Turgut

**Affiliations:** 1Department of Hematology, Çanakkale Mehmet Akif Ersoy State Hospital, 17100 Çanakkale, Türkiye; 2Department of Guidance and Psychological Counseling, Faculty of Education, Amasya University, 05100 Amasya, Türkiye; ertugrulsahin@amasya.edu.tr; 3Department of Guidance and Psychological Counseling, Faculty of Education, Çanakkale Onsekiz Mart University, 17000 Çanakkale, Türkiye; nursel.topkaya@comu.edu.tr; 4Department of Internal Medicine Sciences, Samsun Training and Research Hospital, 55090 İlkadım, Türkiye; oznur.aydin@samsun.edu.tr (Ö.A.); shatun.aktimur@samsun.edu.tr (S.H.A.); 5Department of Hematology, Faculty of Medicine, Ondokuz Mayis University, 55200 Atakum, Türkiye; turgutm@omu.edu.tr

**Keywords:** hematologic cancer patients, DASS-42, DASS-21, validity, reliability, Türkiye

## Abstract

**Background/Objectives**: Patients with hematologic malignancies undergo prolonged, intensive treatments involving frequent hospitalizations and experience debilitating side effects. Consequently, they are at increased risk of developing symptoms of depression, anxiety, and stress, which can undermine their quality of life. However, there is a scarcity of instruments validated for the simultaneous assessment of depression, anxiety, and stress within hematologic cancer populations. The aim of this study is to examine the construct validity, convergent and discriminant validity, and reliability of the Depression Anxiety Stress Scales (DASS-42 and DASS-21) among hematologic cancer patients. **Methods**: We collected data from 452 hematologic cancer patients across three studies. **Results**: Confirmatory factor analyses indicated that the theoretical correlated three-factor model and bifactor model for DASS-42 and DASS-21 responses were sufficient to explain the underlying factor structure of the scales in hematologic cancer patients. However, the bifactor model for DASS-42 and DASS-21 fit better with the data compared to the theoretical correlated three-factor model. In addition, we found the correlated three-factor model and the bifactor structure to exhibit scalar measurement invariance across gender for DASS-42 and DASS-21. DASS-42 and DASS-21 subscales demonstrated weak to strong negative correlations with measures of psychological well-being (happiness, well-being, life satisfaction) and strong positive correlations with measures of similar constructs (depression, anxiety, and stress), thereby supporting their convergent and discriminant validity with theoretically and empirically expected correlations with external criteria. The reliability analyses demonstrated that both DASS-42 and DASS-21 subscales exhibited strong internal consistency and test–retest reliability when assessing symptoms of depression, anxiety, and stress among patients with hematologic malignancies. Moreover, the item-scale convergent and discriminant validity analyses demonstrated that items exhibited higher corrected item–total correlations with their intended subscales than with other subscales across the DASS-42 and DASS-21, providing evidence for the distinct measurement properties of each subscale. **Conclusions**: The findings suggest that the DASS-42 and DASS-21 are psychometrically robust instruments for use in Turkish hematologic cancer patients.

## 1. Introduction

Cancer, characterized by the uncontrolled proliferation of abnormal cells, constitutes a leading global health burden and one of the primary causes of mortality. Globally, in 2022, an estimated 20 million new cancer cases were diagnosed, resulting in 9.7 million deaths [1]. This corresponds to approximately one in six deaths worldwide being attributable to cancer [2]. Furthermore, epidemiological data suggest that approximately 1 in 5 individuals will develop cancer during their lifetime, with a disproportionately higher mortality rate among women, where 1 in 12 is expected to succumb to the disease [1]. In Türkiye, the impact of cancer is also substantial. According to the Republic of Türkiye Ministry of Health General Directorate of Public Health [3], approximately 223.087 new cancer cases were diagnosed in 2019, and malignant and benign neoplasms contributed to one in five deaths in the country during the same year [4]. Hematologic malignancies represent a significant subset of cancer diagnoses, with over 1 million new cases estimated to be reported globally each year [1,5]. In Türkiye, lymphoid and hematopoietic malignant tumors accounted for 1.16% of all deaths in 2023 [6].

Studies consistently demonstrate that cancer patients frequently experience significant psychological distress following diagnosis, including depression, anxiety, and stress [7]. Similar findings have also been reported among patients with hematologic cancers. For example, Clinton-McHarg et al. [8] reported that 27% of patients were diagnosed with anxiety and 17% with depression, with 12% exhibiting comorbid anxiety and depression. Similarly, Bergerot et al. [9] found that 50% of hematologic cancer patients reported significant distress, 47.1% experienced anxiety, and 26% experienced depression. Abuelgasim et al. [10] further corroborated these findings, reporting a 46.5% prevalence of depression, 22.3% prevalence of anxiety, and 18.1% prevalence of comorbid anxiety and depression in patients with hematologic malignancies.

Patients with hematologic malignancies consistently experience diminished quality of life across multiple domains. Specifically, studies have demonstrated that these individuals report lower physical, psychological, emotional, and social well-being compared to the general population [11]. Furthermore, psychological distress, including depression, anxiety, and stress, significantly correlates with reduced quality of life in this patient population [12]. Research has also suggested that the impact of cancer treatment extends beyond immediate medical concerns because patients frequently experience a constellation of long-term symptoms, including pain, sleep disturbances, fatigue, and gastrointestinal issues, along with treatment-related financial burden, which in turn can adversely affect mental health [13]. Untreated depression following a cancer diagnosis has been linked to decreased treatment adherence and poor survival outcomes [14]. Researchers have also identified specific relationships between psychological symptoms and quality of life metrics in the hematologic cancer patient population. For example, Nakano et al. [15] found that anxiety levels correlated with age, nausea, and cognitive function, whereas depression showed a significant association with insomnia. Their research further demonstrated that patients experiencing anxiety and depression exhibited lower physical functioning scores and reported more severe symptoms on standardized quality of life assessments.

Mental health protection in cancer patients is crucial, not only for optimizing treatment outcomes and quality of life but also for reducing health care costs at both systemic and individual levels [16]. The early identification and management of mental health concerns in cancer patients and their families necessitates the use of validated, reliable instruments for measuring depression, anxiety, and stress symptoms. The Depression Anxiety Stress Scales (DASS) is one of the most commonly used instruments to assess negative emotional states in oncology settings. The DASS were designed as a composite measure of negative emotional states, specifically targeting the dimensions of depression, anxiety, and stress. The full DASS consists of 42 items (DASS-42), distributed equally across three scales, each containing 14 items. A shorter version, the DASS-21, includes 21 items with 7 items per scale. The DASS-21 demonstrates psychometric properties comparable to the DASS-42 and is generally preferred in research settings where participant time is limited [17,18]. The depression subscale assesses the presence and severity of depressive symptoms, including self-deprecation, lack of interest in daily activities, inertia, hopelessness, dysphoria, devaluation of life, and anhedonia. The anxiety subscale assesses the presence and severity of anxiety symptoms related to the subjective experience of anxious affect, skeletal musculature effects, situational anxiety, and autonomic arousal. Finally, the stress subscale assesses the presence and severity of stress symptoms, including overreactivity, agitation, nervous arousal, irritability, impatience, and difficulty relaxing [17,18]. Notably, although stress can be conceptualized theoretically as either a stimulus or a response, the DASS stress subscale specifically adopts a response-based framework [19].

Accurate measurement of psychological distress is paramount in cancer research and clinical practice. Although the DASS is not a diagnostic instrument, it serves as a valuable screening and research tool for identifying individuals at risk for psychopathology. However, the construct validity of the DASS remains complex, with studies yielding inconsistent findings across diverse populations. Specifically, researchers have explored various factor structures, including the original three-factor model, a single-factor model, two-factor models, and a bifactor model. Although the three-factor structure is frequently supported [19,20,21,22,23,24,25], some studies suggest that a single-factor [26] or bifactor model [27,28,29] better captures the underlying dimensions of distress, and some studies find that both the three-factor and bifactor models are adequate [30,31]. These discrepancies may arise from several sources, including variations in sample demographics (e.g., age, cultural background), differences in the specific DASS versions used (e.g., DASS-21 vs. DASS-42), variations in the statistical methods employed for factor analysis, and the inherent complexity of the construct of psychological distress itself, which highlight the need for further research to determine the most appropriate factor structure for diverse populations. In cancer populations, the measurement of psychological distress is further complicated by symptoms such as pain, fatigue, mortality concerns, and treatment-related side effects, which can overlap with symptoms of depression and anxiety [32]. Consequently, psychological assessment tools validated in general populations may not perform equivalently in oncology settings.

Hematologic malignancies represent a significant proportion of cancer diagnoses globally, with increasing prevalence particularly in developed nations. This trend has elevated these cancers to a critical public health concern with long-term societal and economic consequences [12]. The pattern is similarly reflected in Türkiye, where hematologic cancers rank among the most frequently diagnosed malignancies [3]. Understanding the prevalence of psychological distress among hematologic cancer patients is fundamental to health care delivery and service development. Such epidemiological data also enable both the implementation of appropriate interventions and the identification of at-risk individuals requiring targeted support [8]. Despite significant medical advancements that have markedly reduced mortality rates in recent years, hematologic malignancies continue to evoke significant fear and uncertainty in affected individuals [9]. This psychological burden places patients at elevated risk for emotional distress, emphasizing the necessity for early detection and intervention strategies [33]. Given the strong psychometric properties of the DASS across psychiatric, medical, and community samples; its theoretically and empirically driven development; its capacity to differentiate among various dimensions of psychological distress; and its utility as a routine clinical outcome measure, the DASS may serve as a valuable instrument for researchers and clinicians working with hematologic cancer patients [34].

A systematic review of literature examining the psychometric properties of both DASS-42 and DASS-21 has identified several significant limitations [31,35]. Research has predominantly focused on DASS-21 psychometric properties, with limited investigation of DASS-42 validity and reliability in clinical populations, particularly cancer patients [19]. Notably, the psychometric properties of either scale remain unexamined specifically within hematologic cancer populations. The generalizability of existing DASS validation studies is further limited by their predominant focus on Western populations, potentially overlooking crucial cultural variations in collectivist societies such as Türkiye. The cancer experience in Turkish society presents unique psychological challenges shaped by traditional family structures, social support systems, and cultural beliefs. Specific cultural practices, such as the perception of cancer as “bad news” and occasional withholding of diagnoses from patients, create distinct challenges for psychological assessment and intervention planning [36]. This cultural context may influence both the manifestation and intensity of psychological distress symptoms, necessitating culturally sensitive assessment tools. The evaluation of DASS-42 and DASS-21 psychometric properties in Turkish hematologic cancer patients is therefore crucial for determining their effectiveness in measuring psychological distress within this specific cultural framework. Furthermore, most existing psychological assessment instruments were developed for general populations, failing to account for the unique symptomatology and experiences associated with hematologic cancers, including treatment-related fatigue, pain, and illness uncertainty. This limitation underscores the importance of validating these scales within specific clinical populations to ensure accurate assessment of psychological needs and appropriate intervention planning.

Turkish hematologic cancer patients face a particularly challenging treatment trajectory, characterized by prolonged and intensive therapies, frequent hospitalizations, and severe side effects that significantly compromise quality of life. Therefore, accurate assessment of their psychological status is crucial for developing culturally tailored and effective intervention programs. However, a significant gap exists in standardized, valid, and reliable measurement tools for psychological distress in this population within Türkiye, hindering both research and clinical practice. To our knowledge, no studies have examined the psychometric properties of the DASS-42 and DASS-21 in Turkish hematologic cancer patients. Addressing this gap is essential because investigating the psychometric properties of the DASS in this specific patient group will provide valuable insights for researchers and clinicians, enhancing their understanding of psychological needs and facilitating targeted intervention design. Consequently, the aim of this study is to examine the construct, convergent and discriminant validity, measurement invariance, and internal consistency and test–retest reliability of the DASS-42 and DASS-21 in Turkish hematologic cancer patients across three studies.


**Study 1: Confirmatory Factor Analysis and Measurement Invariance of the DASS-42 and DASS-21**


## 2. Methods

### 2.1. Participants

The participants consisted of 422 hematologic cancer patients receiving treatment at the hematology clinics of Çanakkale Mehmet Akif Ersoy State Hospital and Samsun Training and Research Hospital. We selected participants using a convenience sampling method based on the following inclusion criteria: (a) confirmed diagnosis of a hematologic malignancy (leukemias, lymphomas, myeloproliferative neoplasms, myelodysplastic syndromes, and plasma cell disorders such as multiple myeloma); (b) age 18 years or older; (c) currently undergoing active treatment (e.g., chemotherapy, immunotherapy, targeted therapy, supportive care); or follow-up care for a hematologic malignancy (d) ability to provide informed consent and complete self-report measures; and (e) fluency in the Turkish language. We excluded participants if they had (a) severe cognitive impairment or inability to comprehend the study procedures, as determined by clinical staff; (b) acute medical emergencies or a critically unstable condition; (c) significant psychiatric comorbidities that would interfere with study participation; or (d) extreme physical debilitation that would prevent completion of study measures.

### 2.2. Measures

DASS-42: We used this measure to assess the presence and severity of depression, anxiety, and stress symptoms in cancer patients [17,18]. Bilgel and Bayram [24] conducted the adaptation of the DASS-42 into Turkish, along with its validity and reliability studies. Şahin et al. [25] later examined the validity and reliability of the DASS-21 using Bilgel and Bayram’s [24] translation. Participants rate each item on a 4-point Likert-type scale ranging from 0 (did not apply to me at all) to 3 (applied to me very much or most of the time), based on the severity of their symptoms of depression, anxiety, or stress over the past week. Scores on the DASS-42 subscales range from 0 to 42, whereas scores on the DASS-21 subscales range from 0 to 21. Higher scores on the depression, anxiety, and stress subscales indicate more severe symptoms of depression, anxiety, and stress, respectively.

### 2.3. Procedure

We collected data face-to-face from hematology patients receiving treatment at the hematology clinics of Samsun Training and Research Hospital and Çanakkale Mehmet Akif Ersoy State Hospital between July 2024 and December 2024. Prior to the study, we obtained the necessary permissions from the Scientific Research and Publication Ethics Committee of Çanakkale Onsekiz Mart University. Additionally, this study adheres to the Ethical Principles for Medical Research Involving Human Participants as outlined in the latest version of the World Medical Association Declaration of Helsinki. We informed participants about the voluntary nature of participation, confidentiality of responses, and their right to withdraw at any time without consequences. All participants provided informed consent and volunteered to participate in the study without receiving any incentives or rewards. Patients at the Çanakkale Mehmet Akif Ersoy State Hospital Hematology Clinic completed the Personal Information Form and DASS-42, whereas those at the Samsun Training and Research Hospital Hematology Clinic completed the full battery of assessments. The administration of data collection instruments required approximately 25 min.

### 2.4. Statistical Analysis

We performed all statistical analyses using SPSS 26 and Mplus 7.2 [37]. We used confirmatory factor analyses (CFA) to determine the underlying factor structure that best represents the responses of the DASS-42 and DASS-21 in hematologic cancer patients. We tested 7 competing models in this study, based on models previously tested in the existing literature and the theoretical framework. A single factor model (Model 1) tests the adequacy of a common underlying factor to account for the pattern of covariance across DASS items and represents a more reasonable baseline comparison model than an independence model, which posits no covariation among DASS items. Correlated 2-factor models load the items of the depression and anxiety subscales (Model 2), depression and stress subscales (Model 3), and anxiety and stress subscales (Model 4) onto a single composite factor to test the adequacy of alternative models that assess the discriminability of depression, anxiety, and stress symptoms. We tested the applicability of the 2-factor models based on previous research demonstrating strong correlations between the depression, anxiety, and stress subscales of the DASS [19,20,21,22,23,24,25,28]. A correlated 3-factor structure (Model 5) represents the theoretical factor structure that Lovibond and Lovibond [18], the developers of the DASS-42 and DASS-21, proposed. Previous studies have provided strong support for a correlated 3-factor structure underlying the DASS-42 and DASS-21 in both clinical and nonclinical samples [19,20,21,22,23,24,25]. Model 6 introduces a second-order general psychological stress latent factor to the theoretical factor structure that Lovibond and Lovibond [18] proposed and tests the extent to which this latent factor explains the depression, anxiety, and stress latent factors. Compared to Model 5, this second-order model is a just-identified model with zero degrees of freedom. As such, its fit to the data is identical to that of Model 5. Finally, Model 7 examines whether the depression, anxiety, and stress items reflect a general psychological stress factor alongside 3 domain-specific factors for depression, anxiety, and stress. The general psychological stress factor evaluates the adequacy of a latent factor underlying all 3 symptoms, providing insight into the shared variance among depression, anxiety, and stress items [27,28,29]. The 3 dimension-specific factors—depression, anxiety, and stress—assess the unique variance associated with each respective dimension.

The sample polychoric correlation matrix served as input, and we used the mean and variance-adjusted weighted least squares (WLSMV) estimation method to test competing models considering the use of ordered polytomous categorical variables with 4 response categories (e.g., Likert-type scales) in this study [38]. We used goodness-of-fit indices to assess how well competing models fit the observed data. In this study, we used the chi-square (χ^2^), comparative fit index (CFI), root mean square error of approximation (RMSEA), and Tucker–Lewis index (TLI) goodness-of-fit indices. These indices are not sensitive to model complexity or sample size and are recommended for use with the WLSMV estimation method [39]. A nonsignificant χ^2^ value indicates that the sample covariance matrix and model-estimated covariance matrix are similar, or the model-estimated covariance matrix reproduces the data well. However, the χ^2^ goodness-of-fit index is highly sensitive to sample size or the number of variables in a model. Thus, a nonsignificant probability value for χ^2^ is generally uncommon when the sample size or the number of variables in a model is large. Therefore, a lower χ^2^ value indicates a better-fitting model without considering its significance.

The CFI compares the specified model with an independence model and measures relative reduction in model misfit comparing the specified model with an independence model. TLI also compares the specified model with an independence model and measures relative reduction in model misfit but contains a penalty for lack of parsimony. The values of CFI and TLI typically range from 0 to 1. CFI and TLI values of 0.95 and above indicate excellent data fit, whereas values ranging from 0.90 to 0.94 indicate good data fit. RMSEA estimates the discrepancy between the model-implied covariance matrix and the observed covariance matrix per degree of freedom. For the RMSEA, values of 0.05 or below are considered evidence of excellent fit, whereas values in the range of 0.06 to 0.08 are considered evidence of acceptable fit. Unlike other fit indices, RMSEA provides 90% confidence intervals, enabling statistical testing of whether the RMSEA value significantly deviates from 0.05. A nonsignificant *p*-value (*p* > 0.05) indicates that the model fits the data well because it suggests no significant difference between the model-implied correlation values and the observed correlation values [40,41,42,43]. We applied a criterion of 0.30 or greater to individual item factor loadings following CFA, with the corresponding *z*-values required to be significant at least *p* < 0.05 [40,43]. We compared the 3-correlated factor model and the bifactor model against alternative models using the chi-square nested difference test, as implemented by the DIFFTEST command in Mplus 7.2 [37].

We conducted multiple group confirmatory factor analyses to examine the measurement invariance of the best-fitting model in DASS-42 and DASS-21 across gender. Because of the well-known limitations of χ^2^ nested difference tests in large samples, we used the changes in CFI, TLI, and RMSEA goodness-of-fit indices to assess measurement invariance. According to Chen [44] and Cheung and Rensvold [45], a decrease in comparative goodness-of-fit indices (e.g., TLI, CFI) of 0.01 or less (ΔCFI or ΔTLI ≤ 0.01) provides reasonable support for a more restrictive model. Additionally, Chen [44] suggested that an increase in RMSEA of 0.015 or less (ΔRMSEA ≤ 0.015) indicates support for the more restrictive model. In goodness-of-fit indices that contain penalties for model misfit (e.g., RMSEA, TLI), it is possible for a more restrictive model to provide a better fit to data than a less restrictive model during invariance testing. In such cases, there is strong support for the more restrictive model [46]. The confirmatory factor analyses and measurement invariance tests were conducted using THETA parameterization in Mplus. Because of the methodological interdependence between thresholds and factor loadings in ordered polytomous categorical variables, constraining only the factor loadings while maintaining stable threshold estimates presents significant challenges during invariance testing, particularly when variables exhibit cross-loadings (i.e., loadings on multiple factors). In such cases, Mplus does not support metric measurement invariance testing for ordered categorical variables with cross-loadings. Consequently, after establishing configural invariance in Model 7, we proceeded directly to testing scalar invariance. This approach aligns with methodologies employed in prior studies examining the measurement invariance of bifactor models for psychological instruments, such as the DASS-21 [27,47].

## 3. Results of Study 1

### 3.1. Sociodemographic Profile of Participants

Table 1 reports the sociodemographic profile of hematologic cancer patients, including gender, age, education level, marital status, employment status, disease duration, and disease type of the participants.

As shown in Table 1, there were 180 (42.7%) female and 242 (57.3%) male hematologic cancer patients. Their ages ranged from 18 to 91 years, with a mean of 58.40 years (*SD* = 14.86). A large proportion of the participants were primary school graduates (*n* = 164; 38.9%) and married (*n* = 325; 77%). Of these participants, 102 (24.2%) were employed, whereas 320 (75.8%) were unemployed. The three most commonly diagnosed cancer types among patients were multiple myeloma (*n* = 114; 27%), non-Hodgkin’s lymphoma (*n* = 92; 21.8%), and essential thrombocytosis (*n* = 30; 7.1%). Disease duration ranged from 1 to 306 months, with a mean of 45.17 months (*SD* = 54.17). Five patients (1.2%) did not report their age, 42 (10%) did not report their hematologic cancer diagnosis, and 42 (9.7%) did not report their disease duration.

### 3.2. Confirmatory Factor Analyses and Measurement Invariance Analyses of the DASS-42 and DASS-21

Table 2 presents the goodness-of-fit indices for the alternative models tested to examine the underlying factor structure of both DASS-42 and DASS-21. Whereas the one-factor and two-factor models demonstrated acceptable fit to the data, the correlated three-factor and bifactor models showed excellent fit to the data. Results of chi-square nested difference tests revealed that both the correlated three-factor and bifactor models provided a significantly better fit to data compared to the one-factor and two-factor models. The item factor loadings for Model 5 of the DASS-42 ranged from 0.69 to 0.91 for the depression latent variable, 0.51 to 0.91 for the anxiety latent variable, and 0.62 to 0.89 for the stress latent variable. For the DASS-21, item factor loadings ranged from 0.76 to 0.91 for the depression latent variable, 0.50 to 0.91 for the anxiety latent variable, and 0.71 to 0.92 for the stress latent variable. In Model 7 for both DASS-42 and DASS-21, all items showed stronger loadings on the general psychological distress factor compared to the specific factors. These findings suggest that both the correlated three-factor model and the bifactor model may be a suitable factor structure for the DASS-42 and DASS-21 among hematologic cancer patients.

Table 3 presents the results of the measurement invariance analysis for the theoretically suggested correlated three-factor model (Model 5) and bifactor model (Model 7) in DASS-42 and DASS-21, conducted by gender. Accordingly, the changes in the fit indices (ΔCFI, ΔTLI, ΔRMSEA) at the configural, metric, and scalar invariance levels for the three-factor model of DASS-42 and DASS-21 are within acceptable limits (ΔCFI ≤ 0.002, ΔTLI ≤ 0.004, ΔRMSEA ≤ 0.002). Similarly, for the bifactor model of DASS-42 and DASS-21, the changes in fit indices for the configural and scalar invariance levels remain within the recommended range (ΔCFI ≤ 0.001, ΔTLI ≤ 0.004, ΔRMSEA ≤ 0.004). These findings suggest that both the theoretically suggested correlated three-factor model and the bifactor model exhibit scalar measurement invariance across gender for DASS-42 and DASS-21.


**Study 2: Convergent and Discriminant Validity of DASS-42 and DASS-21**


## 4. Methods of Study 2

### 4.1. Participants

The participants consisted of 234 hematologic cancer patients receiving treatment at the hematology clinic of Samsun Training and Research Hospital. We selected participants using the convenience sampling method. Among the hematologic cancer patients, 101 (43.2%) were female, and 133 (56.8%) were male. Their ages ranged from 18 to 91 years, with a mean of 57.46 years (*SD* = 16.31). A large proportion of the participants were primary school graduates (*n* = 100; 42.7%) and married (*n* = 182; 77.8%). Of these participants, 47 (20.1%) were employed, whereas 187 (79.9%) were unemployed. The most commonly diagnosed cancer type among cancer patients was multiple myeloma (*n* = 63; 26.9%). Disease duration ranged from 2 to 251 months, with a mean of 40.08 months (*SD* = 45.29). Two patients (0.9%) did not report their age, 39 (16.7%) did not report their hematologic cancer diagnosis, and 36 (15.4%) did not report their disease duration.

### 4.2. Measures

DASS-42: Participants completed the DASS-42, as introduced in the first study.

Patient Health Questionnaire-9 (PHQ-9): We used this measure to assess depressive symptoms that participants experienced over the past 2 weeks [48]. The Turkish version of this scale has demonstrated satisfactory psychometric properties across multiple validity and reliability studies [49,50]. The PHQ-9 consists of nine items, each rated on a 4-point Likert scale ranging from 0 (never) to 3 (almost every day). Total scores can range from 0 to 27, with higher scores reflecting greater symptom severity. A sample item from the scale is “Feeling down, depressed, or hopeless”.

Generalized Anxiety Disorder-7 (GAD-7): We assessed the severity of anxiety symptoms that participants experienced over the past 2 weeks using the GAD-7 [51]. Konkan et al. [52] conducted the Turkish adaptation, validity, and reliability studies of the GAD-7. The GAD-7 consists of seven items, each rated on a 4-point Likert scale ranging from 0 (not at all) to 3 (nearly every day). Total scores range from 0 to 21, with higher scores reflecting greater levels of anxiety symptoms. A sample item from the scale is “Feeling nervous, anxious, or on edge”.

Perceived Stress Scale-10 (PSS-10): We used this scale to assess perceived stress levels that participants experienced over the past month [53]. Eskin et al. [54] conducted the Turkish adaptation, validity, and reliability studies of the PSS-10. The scale consists of 10 items, each rated on a 5-point Likert-type scale ranging from 0 (never) to 4 (very often). Total scores range from 0 to 40, with higher scores indicating greater perceived stress. A sample item from the scale is, “In the last month, how often have you been upset because of something that happened unexpectedly?”

Satisfaction With Life Scale (SWLS): We employed this scale to measure life satisfaction levels among participants [55]. Dağlı and Baysal [56] conducted the Turkish adaptation, validity, and reliability studies of the SWLS. The scale comprises five items, each rated on a 7-point Likert-type scale ranging from 1 (strongly disagree) to 7 (strongly agree). Total scores range from 5 to 35, with higher scores reflecting greater life satisfaction. A sample item from the scale is “I am satisfied with my life”.

Flourishing Scale (FS): We used this scale to measure subjective well-being levels among participants [57]. Telef [58] performed the Turkish adaptation, language, construct, convergent validity, and reliability studies. The scale comprises eight items, each rated on a 7-point Likert-type scale ranging from 1 (strongly disagree) to 7 (strongly agree). Total scores range from 8 to 56, with higher scores reflecting greater subjective well-being. A sample item from the scale is “I am optimistic about my future”.

Single-Item Happiness Scale (SIHS): We employed the SIHS to measure happiness levels among participants [59]. Participants rated their overall happiness by responding to the item, “Taken your life as a whole, how would you rate your happiness?” on a 10-point Likert-type scale ranging from 1 (very unhappy) to 10 (very happy). In addition to substantial evidence supporting the face and content validity of the SIHS, previous investigations of its psychometric properties have revealed strong support for its convergent and divergent validity with statistically significant correlations in the hypothesized directions with measures of depression, anxiety, stress, burnout, self-esteem, and self-efficacy [25,60]. Scores on the SIHS range from 1 to 10, with higher scores reflecting greater happiness.

### 4.3. Procedure

We used the same procedure that we used in the first study.

### 4.4. Statistical Analysis

We conducted all statistical analyses using SPSS 26. We calculated Pearson product-moment correlations to assess the convergent and discriminant validity of the depression, anxiety, and stress subscale scores of the DASS-42 and DASS-21. We conducted a series of Steiger’s z-tests for dependent samples to examine whether there were significant differences in the correlation values between the DASS-42 and DASS-21 depression, anxiety, and stress subscales with variables measuring similar (depression, anxiety, and stress) and different (life satisfaction, subjective well-being, and happiness) constructs. We used Cohen’s [61] effect size classifications to interpret correlation coefficients. We set statistical significance at *p* < 0.05 for all statistical analyses.

## 5. Results of Study 2

Table 4 presents the means, standard deviations, and Pearson product-moment correlation coefficients between the DASS-42 and DASS-21 depression, anxiety, and stress subscales and convergent and divergent validity measures.

As shown in Table 4, the DASS-42 and DASS-21 depression, anxiety, and stress subscales strongly and positively correlated with PHQ-9, GAD-7, and PSS-10 scores. Although the DASS-42 and DASS-21 depression subscale scores were highly negatively correlated with SWLS and SIHS scores, they were moderately negatively correlated with FS scores. Furthermore, the DASS-42 and DASS-21 anxiety subscale scores were moderately and negatively correlated with the FS, SWLS, and SIHS scores. In contrast, the DASS-42 and DASS-21 stress subscale scores were moderately negatively correlated with the SWLS and SIHS scores and weakly negatively correlated with FS scores.

To assess the comparability of convergent and discriminant validity between the DASS-21 and DASS-42 subscales, we employed Steiger’s z-tests to compare the strength of correlations between each DASS subscale (depression, anxiety, and stress for both DASS versions) and various criterion variables (PHQ-9, GAD-7, PSS-10, SWLS, and SIHS). Table 5 presents the results of Steiger’s dependent samples *z*-tests.

As shown in Table 5, Steiger’s z-test results revealed that only 3 out of 18 comparisons were statistically significant. The comparison of the DASS-42 and DASS-21 depression, anxiety, and stress scales with the criterion variables revealed that 83.33% of the correlation coefficients did not show a statistically significant difference. The analysis of the significantly different correlation coefficients showed that the DASS-42 depression and anxiety subscales were more strongly correlated with the PHQ-9 depression scale scores than the corresponding DASS-21 depression and anxiety subscales. Additionally, the DASS-42 stress subscale showed a stronger relationship with the GAD-7 anxiety scores than the DASS-21 stress subscale. Overall, the strength of relationships between the DASS-42 and DASS-21 subscales and measures of convergent and divergent validity were generally consistent across the 2 scales.


**Study 3: Internal Consistency and Test–Retest Reliability of the DASS-42 and DASS-21**


## 6. Methods of Study 3

### 6.1. Participants

We conducted internal consistency reliability analyses of the DASS-42 and DASS-21 using data from the first sample group. To assess test–retest reliability, we collected data from 34 cancer patients at 1-month intervals. Of these, 10 (29.4%) were female and 24 (70.6%) were male. The age of participants ranged from 21 to 85 years, with a mean age of 56.65 years (*SD* = 13.69). Of these participants, 12 (35.3%) were employed, whereas 22 (64.7%) were unemployed. The most common medical diagnosis was multiple myeloma (*n* = 14, 41.2%).

### 6.2. Measures

DASS-42: Participants completed the DASS-42, which was introduced in Study 1.

### 6.3. Procedure

We followed the same procedure used in Study 1.

### 6.4. Statistical Analysis

We conducted all statistical analyses using SPSS 26. We performed a series of dependent samples *t*-tests to assess whether there was a statistically significant change in the DASS-42 and DASS-21 depression, anxiety, and stress subscale scores over a 1-month time interval. The test–retest reliability of the DASS-42 and DASS-21 depression, anxiety, and stress subscales was evaluated using both Pearson product–moment correlations and intraclass correlation coefficients (ICC; two-way mixed, absolute agreement). We assessed internal consistency reliabilities of DASS-42 and DASS-21 by calculating corrected item–total correlations and Cronbach’s alpha coefficients. We employed the method that Sinclair et al. [62] proposed to examine item-scale convergent and discriminant validity. We used Cohen’s [61] effect size classifications for interpreting correlation values, whereas we followed the guidelines that Koo and Li [63] suggested for interpreting ICC values. As a general rule, corrected item–total correlations of 0.30 and above are considered adequate [43,64]. Additionally, scales with a reliability coefficient above 0.70 are considered adequate for screening and research purposes [64,65,66]. To further support the evidence of reliability, we examined floor and ceiling effects. The percentage of respondents achieving the highest possible total score indicates the ceiling effect, whereas the percentage of respondents with the lowest possible total score reflects the floor effect [66]. We set statistical significance at *p* < 0.05 for all statistical analyses.

## 7. Results of Study 3

### 7.1. Results of Test–Retest Reliability of the DASS-42 and DASS-21

Table 6 presents the results of dependent samples *t*-tests examining changes in DASS-42 and DASS-21 depression, anxiety, and stress scores over a 1-month interval.

As shown in Table 6, results of the dependent samples *t*-test revealed no significant difference between the mean depression, anxiety, and stress scores of the DASS-42 and DASS-21 administered to cancer patients 1 month apart.

Table 7 presents the zero-order correlations and intraclass correlation coefficients (ICCs) for the DASS-42 and DASS-21 depression, anxiety, and stress subscales, assessing test–retest reliability over a 1-month interval. As shown in Table 6, The Pearson correlation coefficients were 0.83 (depression), 0.88 (anxiety), and 0.81 (stress) for the DASS-42 and 0.83 (depression), 0.83 (anxiety), and 0.74 (stress) for the DASS-21, indicating very high test–retest reliability values for the DASS-42 and DASS-21 subscales over a 1-month interval. The intraclass correlation analysis also revealed good reliability for all subscales of the DASS-42 as well as for the depression and anxiety subscales of the DASS-21. The DASS-21 stress subscale exhibited moderate reliability based on ICC values.

### 7.2. Results of Item Analyses of the DASS-42 and DASS-21

Table 8 presents corrected item–total correlations, item-scale convergent and discriminant validity results, Cronbach’s alpha coefficients, and ceiling and floor effect analyses for the DASS-42 and DASS-21.

As shown in Table 8, the item–total correlation values for the DASS-42 depression scale ranged from 0.57 to 0.83, the anxiety scale from 0.43 to 0.73, and the stress scale from 0.53 to 0.75. In contrast, for the DASS-21, the item–total correlation values for the depression scale ranged from 0.69 to 0.81, the anxiety scale from 0.42 to 0.74, and the stress scale from 0.55 to 0.69. In the DASS-42, 85.7% (12/14) of the depression subscale items, 71.4% (10/14) of the anxiety subscale items, and 78.6% (11/14) of the stress subscale items showed a stronger correlation with their respective subscales. In the DASS-42 depression subscale, item 13 (“I felt sad and depressed”) showed a stronger corrected item–total correlation with the stress subscale, whereas item 42 (“I found it difficult to work up the initiative to do things”) showed a stronger corrected item–total correlation with the anxiety subscale. In the DASS-42 anxiety subscale, item 9 (“I found myself in situations that made me so anxious I was most relieved when they ended”) and item 28 (“I felt I was close to panic”) showed a stronger corrected item–total correlation with the stress subscale, whereas item 30 (“I feared that I would be ‘thrown’ by some trivial but unfamiliar task”) and item 36 (“I felt terrified”) showed a stronger corrected item–total correlation with the depression subscale. In the DASS-42 stress subscale, item 8 (“I found it difficult to relax”) and item 22 (“I found it hard to wind down”) showed stronger corrected item–total correlation values with the depression subscale, whereas item 39 (“I found myself getting agitated”) showed a stronger corrected item–total correlation value with the anxiety subscale.

In the DASS-21, 85.7% (6/7) of the depression subscale items, 57.1% (4/7) of the anxiety subscale items, and 71.4% (5/7) of the stress subscale items showed a stronger correlation with their respective subscales. Whereas item 5 (“I found it difficult to work up the initiative to do things”) of the DASS-21 depression subscale showed a stronger corrected item–total correlation with the anxiety subscale, item 15 (“I felt I was close to panic”) of the DASS-21 anxiety subscale showed a stronger corrected item–total correlation with the stress subscale, and item 9 (“I was worried about situations in which I might panic and make a fool of myself”) showed a stronger corrected item–total correlation with the depression subscale. In the DASS-21 stress subscale, item 11 (“I found myself getting agitated”) showed a stronger corrected item–total correlation with the anxiety subscale, whereas item 1 (“I found it hard to wind down”) showed a stronger corrected item–total correlation with the depression subscale. The percentage of respondents who obtained the highest possible total score ranged from 0.0% to 0.7% in the DASS-42 and from 0.0% to 4.3% in the DASS-21, whereas the percentage of respondents with the lowest possible total score ranged from 3.3% to 11.4% in the DASS-42 and from 8.3% to 18.2% in the DASS-21.

## 8. Discussion

In this study, we examined the validity and reliability of the DASS-42 and DASS-21 among hematologic cancer patients. We tested seven alternative models to assess the construct validity of the DASS-42 and DASS-21, and we found that both the theoretically proposed correlated three-factor structure and the bifactor structure provided a better fit to the data compared to the other competing models. However, the bifactor model demonstrated a slightly better fit to the data compared to the correlated three-factor structure. Model parameter estimates, including factor loadings and standard errors, were adequate for both models. These findings support the correlated three-factor structure that Lovibond and Lovibond [18] proposed. The findings align with those of previous studies using the DASS-21 in both clinical and nonclinical samples, which have shown that the scale fits well with both the correlated three-factor structure and the bifactor model [30,31]. For example, the meta-confirmatory factor analysis by Yeung et al. [31] demonstrated that both the correlated three-factor structure and the bifactor model provided a good fit to the DASS-21 data, although the bifactor model showed a slightly better fit. Consistently, in a study involving university students from eight countries, Zanon et al. [30] found that both the correlated three-factor structure and the bifactor model provided a good fit to the DASS-21 data across all countries. Furthermore, the researchers reported that, based on model-based validity and reliability analyses, a significant portion of the systematic variance in DASS-21 depression, anxiety, and stress scores could be attributed to individual differences in the general psychological distress factor, suggesting that evaluations based solely on subscale scores may be misleading. Similarly, Lee et al. [35] conducted a comprehensive systematic review of the psychometric properties of the DASS-21 and concluded that the bifactor model consistently demonstrated the strongest evidence of validity and reliability across clinical and nonclinical samples. Consistent with existing literature on the DASS-21, the present study supports its validity among cancer patients, extending its generalizability to hematologic cancer patients and confirming its utility alongside the DASS-42.

The bifactor model’s slightly better fit in the present study highlights the interconnected nature of depression, anxiety, and stress symptoms, which aligns with the theoretical foundation of the DASS. Although DASS-42 and DASS-21 were designed to maximally differentiate depression, anxiety, and stress symptoms based on a tripartite model of anxiety and depression, a lack of specificity may be expected in these 3 emotional states among hematologic cancer patients. Empirical studies have found that mood and anxiety disorder symptoms are very strongly correlated with each other among adults, and adults who are diagnosed with mood or anxiety disorders in their lifetime have an increased risk for subsequently developing the other disorder [67,68]. Tiller [69] posited that depression and anxiety disorders exhibit commonality in terms of general, cognitive, emotional, psychological, and physical symptoms. It is also possible to see this overlap in the DASS-21 items. All 3 subscales of the DASS-21 assess negative affect (emotional stress) such as feelings of dysphoria and worthlessness in the depression subscale, anxious affect and situational anxiety in the anxiety subscale, and nervous arousal and irritability in the stress subscale [17,18]. Depression, anxiety, and stress symptoms often manifest themselves as changes in cognitive processes and behaviors [70,71]. For example, these changes were measured in the depression subscale with anhedonia (loss of interest or pleasure in daily activities) and inertia (feeling slowed down or lacking energy), which can also be related to anxiety and stress symptoms. The stress subscale includes items related to irritability, impatience, and overreaction, which can be associated with both anxiety and depression symptoms in adult individuals [72]. Moreover, physiological responses to stressors can be triggered by anxiety (e.g., autonomic arousal, skeletal musculature effects) and stress (e.g., nervous arousal, agitation) symptoms, which in turn lead to increases in depressive symptoms [73]. Given the interconnected nature of depression, anxiety, and stress symptoms in daily life, individuals can experience difficulty differentiating and separately conceptualizing these symptoms. This difficulty can occur because these symptoms often reflect a general psychological distress factor, as demonstrated by the bifactor model of DASS-42 and DASS-21 in hematologic cancer patients.

Multiple group confirmatory factor analyses revealed that both the DASS-42 and DASS-21 demonstrated configural, metric, and scalar invariance across gender for the correlated three-factor and bifactor model. These results align with prior research showing that the correlated three-factor model [21,22,23,26] and the bifactor model [29] of the DASS-21 exhibit measurement invariance across gender in nonclinical samples. The current findings extend this evidence to the DASS-42 and DASS-21 in clinical samples, specifically hematologic cancer patients, confirming their configural, metric, and scalar invariance across gender. The configural invariance of the DASS-42 and DASS-21 suggests that women and men conceptualize depression, anxiety, stress, or general psychological distress similarly. In other words, the overall latent factor structure of the DASS-42 and DASS-21 (whether the correlated three-factor or bifactor model) remains consistent across women and men with hematologic cancer. The metric invariance further suggests that the strength of the relationship between each DASS item and its respective latent construct (depression, anxiety, stress, or general psychological distress) is comparable for women and men. In practical terms, this means that changes in the latent constructs are reflected in changes in observed DASS scores in a similar manner for both genders. The scalar invariance of both models across both versions indicates that individuals, regardless of gender, have the same expected response at the same absolute latent level of depression, anxiety, stress, or psychological distress. This suggests that male and female hematologic cancer patients interpret the response categories similarly, and any observed differences in item responses can be attributed solely to actual differences in latent factor means rather than measurement bias [40,42]. Because the observed gender differences in scores reflect actual differences in depression, anxiety, and stress levels rather than measurement artifacts, mean scores on the DASS-42 and DASS-21 subscales can be validly compared between male and female hematologic cancer patients [40,42].

We assessed the convergent and discriminant validity of the DASS-42 and DASS-21 by examining their correlations with measures of psychological well-being (e.g., happiness, well-being, life satisfaction) and similar constructs (e.g., depression, anxiety, stress). As expected, the DASS-42 and DASS-21 subscales demonstrated strong convergent and discriminant validity by exhibiting high positive correlations with measures of depression, anxiety, and stress and moderate to high negative correlations with measures of psychological well-being. These findings are consistent with previous research supporting the convergent and discriminant validity of DASS [19,20,25,29,74]. For example, Bengwasan et al. [20] demonstrated strong positive correlations between DASS-21 subscales and both PHQ-9 and GAD-7 scores, alongside moderate to high negative correlations with FS scores. Similarly, Lee and Kim [29] found moderate to high positive correlations between DASS-21 subscales and PHQ-9, GAD-7, and PSS-10 scores. In the context of cancer patients, Soria-Reyes et al. [19] observed moderate negative correlations between DASS-21 subscales and SWLS scores. Şahin et al. [25] reported moderate to high negative correlations between DASS-21 subscales and SIHS scores in an adult sample. Moreover, results of Steiger’s z-test analyses comparing DASS-42 and DASS-21 revealed significant differences in only 3 out of 18 comparisons with criterion variables (PHQ-9, GAD-7, PSS-10, SWLS, FS, SIHS). However, 15 out of 18 comparisons (83.33%) were not significant, indicating that the strengths of correlations were generally statistically equivalent. In practical terms, these findings suggest that the DASS-21 subscales demonstrate very similar levels of convergent and discriminant validity to the DASS-42 subscales when related to measures of depression, anxiety, stress, and well-being indicators (life satisfaction, well-being, happiness). The shorter DASS-21 appears to capture the same validity relationships as the DASS-42 in most cases.

Test–retest reliability, item analysis, internal consistency, and floor and ceiling effect analyses also supported the reliability of the DASS-42 and DASS-21. Because test–retest reliability analyses show that the DASS-42 and DASS-21 depression, anxiety, and stress subscales exhibit good temporal stability over a 1-month interval and produce similar and relatively stable scores over time [61,63], these subscales can be used to examine differences in depression, anxiety, and stress scores across and within groups over time as well as to assess the effectiveness of treatments and interventions. Item analyses of the DASS-42 and DASS-21 revealed moderate to high corrected item–total correlations across the depression, anxiety, and stress subscales. These correlations indicate that individual item scores demonstrated strong associations with their respective subscale total scores, suggesting that items effectively discriminate between individuals with varying levels of depression, anxiety, and stress symptoms [64,65]. Cronbach’s alpha reliability values indicated strong internal consistency among the items of the DASS-42 and DASS-21 depression, anxiety, and stress subscales, with values ranging from high to very high, demonstrating that both scales exhibit strong reliability and are suitable for screening and research purposes [43,64,65]. These findings align with previous research consistently demonstrating high reliability coefficients for both the DASS-42 and DASS-21 across diverse populations and contexts [18,19,20,23,24,29,74].

Furthermore, we examined the item-scale convergent and discriminant validity of the DASS-42 and DASS-21 subscales using the method that Sinclair et al. [62] proposed. The results indicated that 85.7% (12/14) of the DASS-42 depression subscale items, 71.4% (10/14) of the anxiety subscale items, and 78.6% (11/14) of the stress subscale items had stronger correlations with their respective subscales. For the DASS-21, 85.7% (6/7) of the depression subscale items, 57.1% (4/7) of the anxiety subscale items, and 71.4% (5/7) of the stress subscale items showed stronger correlations with their own subscales. These findings are consistent with Sinclair et al.’s [62] results regarding DASS-21’s item-scale convergent and discriminant validity while extending the evidence to DASS-42. These findings also support Watson and Clark’s [75] tripartite model of anxiety and depression, which conceptualizes depression, anxiety, and stress as interrelated but distinct constructs. Additionally, floor and ceiling effect analyses revealed that the percentage of participants scoring at the extreme ends was generally below the 15% threshold [66], with the exception of the DASS-21 depression subscale. This general absence of floor and ceiling effects strengthens evidence for the reliability, content validity, and responsiveness of both the DASS-42 and DASS-21 subscales [66].

### 8.1. Practical Implications

The results of this study have several practical implications for clinical practice and research with hematologic cancer patients. The results demonstrate that the DASS-42 and DASS-21 are reliable and valid instruments for assessing depression, anxiety, and stress levels in hematologic cancer patients, offering clinicians and researchers a valuable tool for patient assessment. Clinicians can use these scales to monitor changes in depression, anxiety, and stress levels over time, allowing them to track patient progress and adjust treatment plans accordingly. The bifactor model findings, which highlight the interconnected nature of depression, anxiety, and stress, suggest that clinicians should implement integrated treatment approaches rather than addressing these conditions in isolation. This interconnectedness supports the use of transdiagnostic interventions such as acceptance and commitment therapy or mindfulness-based cognitive therapy, which target underlying psychological mechanisms shared across these symptom domains. For medication management, these findings suggest psychiatrists should consider pharmacotherapies with broader effects across symptom domains rather than highly targeted medications for single conditions.

The measurement invariance of the DASS-42 and DASS-21 across gender suggests that these scales can be used to compare depression, anxiety, and stress levels between male and female hematologic cancer patients, facilitating the development of gender-sensitive intervention plans and implementation of targeted support strategies based on gender-specific needs. Furthermore, the scales demonstrate good test–retest reliability, making them suitable for use in treatment processes to monitor changes in symptoms of psychological distress throughout the course of cancer treatment and evaluate the effectiveness of intervention programs. The finding that the DASS-21 shows similar psychometric properties to the DASS-42 as a short form suggests that the shorter version may be preferred in time-constrained clinical settings or when patient fatigue is a significant factor, such as during chemotherapy sessions or presurgical evaluations. This is particularly relevant given the increased prevalence of fatigue in hematologic cancer patients, as previous studies [76] have documented. Finally, the general absence of floor and ceiling effects in the scales suggests that they can be used to assess patients with varying levels of symptom severity, which is crucial for monitoring patients with both mild and severe symptoms. Overall, these validated instruments provide oncology teams with reliable tools to inform psychological intervention planning, evaluate treatment effectiveness, and tailor supportive care approaches to the unique needs of hematologic cancer patients throughout their treatment journey.

### 8.2. Limitations

This study has several limitations that should be considered when interpreting the findings. First, we collected the data from only two hospitals (Samsun Training and Research Hospital and Çanakkale Mehmet Akif Ersoy State Hospital), which limits the generalizability of the findings to all hematologic cancer patients in Türkiye. Additionally, the external validity of the findings is limited, not only because the hematologic cancer patients studied represent only a small minority of the global cancer patient population but also because the interpretation and expression of symptoms of depression, anxiety, and stress may vary significantly across different groups of cancer patients.

Although the DASS-42 and DASS-21 exhibit robust psychometric properties for assessing depression, anxiety, and stress, their scope is inherently limited in capturing the full spectrum of psychological distress that hematologic cancer patients experience. These instruments do not specifically assess cancer-related concerns such as fear of recurrence, existential distress, body image issues, or treatment-specific anxieties, which are often prominent in this population [77]. Additionally, they do not measure other relevant mental health constructs, such as post-traumatic stress symptoms, complicated grief reactions, or adjustment disorders, which may be prevalent among cancer patients facing life-threatening illness [78,79]. Furthermore, these scales may not adequately capture culturally specific expressions of distress that could be significant in Turkish populations. Future research should consider supplementing DASS assessments with cancer-specific psychological measures to achieve a more comprehensive understanding of psychological functioning in hematologic cancer patients.

Another limitation pertains to the demographic variations within the sample, particularly regarding age, which ranged from 18 to 91 years across the studies. This wide age range introduces potential heterogeneity that may influence the interpretation of the results. For example, younger patients might experience distress related to disruptions in developmental milestones (e.g., education, career, family planning), whereas older patients might face distress tied to comorbidities, reduced physical resilience, or existential concerns about mortality. Such age-related differences could affect how depression, anxiety, and stress are experienced and reported on the DASS-42 and DASS-21, potentially affecting the scales’ sensitivity and specificity across age groups. Although the sample size was sufficient to support the psychometric analyses, the study did not explicitly examine age as a moderating factor, limiting our understanding of its influence on the scales’ performance. Future research should explore age-stratified analyses or measurement invariance testing across age groups to clarify these potential effects.

The reliance on self-report data collection tools introduces additional limitations, such as the potential for social desirability bias or recall bias to affect the research results. We also did not account for external factors (e.g., socioeconomic status, psychotropic medication use) that may influence symptom intensity and lead to systematic variation in item responses because these factors were not attributed to the DASS latent constructs. Although the large sample size likely mitigates the impact of these unmeasured factors [40,42], future research should explicitly include these variables in the study design to explore their potential influence on DASS scores. In addition, future research should investigate the predictive validity of the DASS-42 and DASS-21 by examining their ability to predict clinically relevant outcomes, such as treatment response, disease progression, or quality of life, over time. Researchers could also explore the predictive validity of specific factors related to depression, anxiety, and stress beyond the general stress factor using external criteria.

## 9. Conclusions

In the present study, we evaluated the psychometric properties of the DASS-42 and DASS-21, including construct validity, convergent and discriminant validity, internal consistency, and test–retest reliability. The results demonstrated that both the theoretical three-factor model and the bifactor model provided a good fit for the factor structure of the DASS-42 and DASS-21. Furthermore, scalar measurement invariance across gender was established for both the correlated three-factor and bifactor models, indicating that the scales function similarly for male and female participants. The DASS-42 and DASS-21 also demonstrated convergent validity through strong positive correlations with measures of depression, anxiety, and stress and discriminant validity via low to moderate negative correlations with measures of happiness, well-being, and life satisfaction. The reliability analyses further confirmed that both the DASS-42 and DASS-21 are internally consistent and produce similar and relatively stable scores over time, making them reliable tools for assessing symptoms of depression, anxiety, and stress in Turkish hematologic cancer patients. Overall, this study demonstrates that the DASS-42 and DASS-21 are psychometrically sound instruments suitable for use among Turkish hematologic cancer patients. Healthcare professionals can confidently use these instruments to screen for and monitor psychological distress in this population during patient evaluation and intervention planning.

## Figures and Tables

**Table 1 jcm-14-02097-t001:** Sociodemographic profile of study participants.

Variables	Values
Sex, *n* (%)	
Female	180 (42.7)
Male	242 (57.3)
Age	
*M* [Min, Max.]	58.40 [18, 91]
Education Level, *n* (%)	
Literate	16 (3.8)
Primary school	164 (38.9)
Secondary school	43 (10.2)
High school	94 (22.3)
Associate degree	29 (6.9)
Bachelor’s degree or above	76 (18.0)
Marital Status, *n* (%)	
Single	97 (23.0)
Married	325 (77.0)
Employment Status, *n* (%)	
Employed	102 (24.2)
Unemployed	320 (75.8)
Disease Duration	
*M* [Min, Max.]	45.17 [1, 306]
Disease Type, *n* (%)	
Chronic myeloid leukemia	22 (5.7)
Multiple myeloma	114 (27)
Acute myeloid leukemia	27 (6.4)
Essential thrombocytosis	30 (7.1)
Polycythemia vera	21 (5.0)
Acute lymphoblastic leukemia	8 (1.9)
Hodgkin’s lymphoma	25 (5.9)
Chronic lymphocytic leukemia	18 (4.3)
Myelofibrosis	11 (2.6)
Myelodysplastic syndrome	12 (2.8)
Non-Hodgkin’s lymphoma	92 (21.8)
Missing	42 (10.0)

*Note*: *N* = 422.

**Table 2 jcm-14-02097-t002:** Results of DAS-42 and DASS-21 confirmatory factor analyses.

	χ^2^	*df*	CFI	TLI	RMSEA	*p*	RMSEA 90%		Model Summary			
									F1	F2	F3	F4
							LB	UB	λ	λ	λ	λ
DASS-42												
Model 1	2336.899	819	0.942	0.939	0.066	0.001 ***	0.063	0.069	0.50–0.90			
Model 2	2124.743	818	0.950	0.948	0.062	0.001 ***	0.058	0.065	0.53–0.91	0.62–0.89		
Model 3	2264.911	818	0.945	0.942	0.065	0.001 ***	0.062	0.068	0.61–0.91	0.51–0.91		
Model 4	2239.156	818	0.946	0.943	0.064	0.001 ***	0.061	0.067	0.69–0.92	0.50–0.90		
Model 5	2076.067	816	0.952	0.949	0.060	0.001 ***	0.057	0.064	0.69–0.91	0.51–0.91	0.62–0.89	
Model 6	2076.067	816	0.952	0.949	0.060	0.001 ***	0.057	0.064	0.69–0.91	0.51–0.91	0.62–0.89	0.93–0.98
Model 7	1545.531	777	0.971	0.968	0.048	0.768	0.045	0.052	−0.14–0.37	−0.23–0.50	−0.02–0.51	0.49–0.91
DASS-21												
Model 1	605.904	189	0.969	0.965	0.072	0.001 ***	0.066	0.079	0.49–0.90			
Model 2	580.536	188	0.971	0.967	0.070	0.001 ***	0.064	0.077	0.50–0.90	0.70–0.92		
Model 3	583.126	188	0.970	0.967	0.071	0.001 ***	0.064	0.077	0.69–0.90	0.50–0.91		
Model 4	578.366	188	0.971	0.967	0.070	0.001 ***	0.064	0.077	0.76–0.91	0.50–0.89		
Model 5	556.316	186	0.972	0.969	0.069	0.001 ***	0.062	0.075	0.76–0.91	0.50–0.91	0.71–0.92	
Model 6	556.316	186	0.972	0.969	0.069	0.001 ***	0.062	0.075	0.76–0.91	0.50–0.91	0.71–0.92	0.97–0.98
Model 7	372.427	168	0.985	0.981	0.054	0.198	0.046	0.061	0.01–0.35	−0.11–0.50	−0.05–0.49	0.49–0.90

*Note:* LB = lower bound, UB = upper bound, λ = item factor loading. In Model 1, we loaded all items onto a single latent variable, the psychological stress latent factor. In Model 2, items from the depression and anxiety dimensions form the first latent variable (F1), whereas the stress items form the second latent variable. In Model 3, items from the depression and stress dimensions form the first latent variable (F1), whereas anxiety items form the second latent variable. In Model 4, items from the depression dimension form the first latent variable (F1), whereas items from the anxiety and stress dimensions form the second latent variable (F2). Model 5 represents the theoretical factor structure that Lovibond and Lovibond [18] proposed, where items in the depression dimension form the depression latent variable (F1), items in the anxiety dimension form the anxiety latent variable (F2), and items in the stress dimension form the stress latent variable (F3). Model 6, in addition to Model 5, examines the extent to which the second-order psychological stress latent variable (F4) can be explained by the depression (F1), anxiety (F2), and stress (F3) latent variables. Finally, Model 7 tests whether the depression (F1), anxiety (F2), and stress (F3) items represent a general psychological stress factor as well as 3 domain-specific factors for depression, anxiety, and stress. Except for Model 7, all factor loadings were significant at least at the *p* < 0.001 level in the tested models. In Model 7, however, values of |0.10| and above were considered significant at the *p* < 0.05 level; *p* < 0.001 ***.

**Table 3 jcm-14-02097-t003:** Results of measurement invariance analyses of the DASS-42 and DASS-21 across gender.

Measurement Invariance	χ^2^	*df*	CFI	TLI	RMSEA	ΔCFI	ΔTLI	ΔRMSEA
DASS-42								
Model 5								
Configural	2875.547	1632	0.956	0.954	0.060			
Metric	2910.511	1671	0.956	0.955	0.059	0.000	0.001	−0.001
Scalar	2939.517	1752	0.958	0.959	0.057	0.002	0.004	−0.002
Model 7								
Configural	2368.706	1554	0.971	0.968	0.050			
Scalar	2494.245	1714	0.972	0.972	0.046	0.001	−0.004	0.004
DASS-21								
Model 5								
Configural	743.056	372	0.973	0.969	0.069			
Metric	762.232	390	0.973	0.971	0.067	0.000	0.002	−0.002
Scalar	784.361	429	0.974	0.975	0.063	0.001	0.004	−0.004
Model 7								
Configural	558.504	336	0.984	0.980	0.056			
Scalar	630.877	412	0.984	0.984	0.050	0.000	0.004	−0.006

**Table 4 jcm-14-02097-t004:** Results of the Pearson product-moment correlation analyses.

Scales	1	2	3	4	5	6	7	8	9	10	11	12
DASS-42												
1. Depression												
2. Anxiety	0.84											
3. Stress	0.81	0.75										
DASS-21												
4. Depression	0.98	0.83	0.78									
5. Anxiety	0.80	0.96	0.73	0.78								
6. Stress	0.81	0.75	0.96	0.77	0.71							
Convergent validity												
7. PHQ-9	0.80	0.79	0.70	0.77	0.75	0.69						
8. GAD-7	0.77	0.73	0.81	0.76	0.71	0.76	0.74					
9. PSS-10	0.66	0.59	0.62	0.64	0.58	0.61	0.65	0.68				
Discriminant validity												
10. SWLS	−0.54	−0.46	−0.41	−0.55	−0.44	−0.39	−0.46	−0.44	−0.53			
11. FS	−0.40	−0.34	−0.27	−0.41	−0.33	−0.28	−0.35	−0.32	−0.48	0.65		
12. SIHS	−0.55	−0.45	−0.45	−0.55	−0.42	−0.45	−0.49	−0.46	−0.44	0.54	0.45	
*M*	7.71	8.52	11.38	3.71	4.24	4.89	6.28	4.41	15.56	24.06	42.76	7.27
*SD*	7.35	6.50	8.38	3.62	3.48	3.98	5.36	4.82	6.70	7.47	11.59	2.14
Skewness	1.19	0.98	0.80	1.12	0.89	0.86	0.89	1.07	−0.03	−1.15	−0.70	−0.61
Kurtosis	1.17	0.87	0.07	0.90	0.24	0.10	0.09	0.48	−0.40	0.73	−0.26	0.08
Cronbach’s alpha (α)	0.93	0.88	0.93	0.85	0.79	0.85	0.86	0.92	0.79	0.92	0.95	-

*Note*: *N* = 234. All correlation values are significant at least at the *p* < 0.001 level.

**Table 5 jcm-14-02097-t005:** Results of dependent samples Steiger’s *z*-tests.

Criterion Variable	DASS-42 Subscale	DASS-21 Subscale	*z*	*p*
PHQ-9	Depression	Depression	3.18	0.001 **
GAD-7	Depression	Depression	1.51	0.130
PSS-10	Depression	Depression	1.86	0.063
SWLS	Depression	Depression	1.33	0.184
FS	Depression	Depression	1.14	0.256
SIHS	Depression	Depression	0.18	0.859
PHQ-9	Anxiety	Anxiety	3.24	0.001 **
GAD-7	Anxiety	Anxiety	1.56	0.119
PSS-10	Anxiety	Anxiety	1.05	0.292
SWLS	Anxiety	Anxiety	−1.49	0.136
FS	Anxiety	Anxiety	0.17	0.865
SIHS	Anxiety	Anxiety	−1.89	0.058
PHQ-9	Stress	Stress	0.88	0.378
GAD-7	Stress	Stress	4.66	0.001 ***
PSS-10	Stress	Stress	0.67	0.502
SWLS	Stress	Stress	−1.32	0.187
FS	Stress	Stress	0.11	0.913
SIHS	Stress	Stress	0.18	0.860

*Note: p* < 0.01 **, *p* < 0.001 ***.

**Table 6 jcm-14-02097-t006:** Results of the dependent samples *t*-tests.

Scale	*n*	*M*	*SD*	*df*	*t*	*p*	*d*
DASS-42 Depression							
First administration	34	8.12	6.31	33	0.37	0.713	0.06
Second administration	34	7.85	7.46				
DASS-42 Anxiety							
First administration	34	8.68	5.15	33	1.82	0.078	0.31
Second administration	34	7.82	5.72				
DASS-42 Stress							
First administration	34	12.29	6.86	33	0.08	0.938	0.01
Second administration	34	12.24	7.15				
DASS-21 Depression							
First administration	34	3.74	3.17	33	−0.30	0.765	−0.05
Second administration	34	3.85	4.09				
DASS-21 Anxiety							
First administration	34	4.12	2.52	33	0.90	0.374	0.15
Second administration	34	3.85	3.05				
DASS-21 Stress							
First administration	34	5.32	3.24	33	−0.29	0.776	0.13
Second administration	34	5.44	3.42				

**Table 7 jcm-14-02097-t007:** Zero-order correlations and ICCs for test–retest reliability of DASS-42 and DASS-21.

Scale	*n*	*r*	ICC	ICC 95% Confidence Interval	
				Lower Bound	Upper Bound
DASS-42 Depression					
First administration	34	0.83	0.82	0.67	0.91
Second administration	34				
DASS-42 Anxiety					
First administration	34	0.88	0.87	0.75	0.93
Second administration	34				
DASS-42 Stress					
First administration	34	0.81	0.81	0.66	0.90
Second administration	34				
DASS-21 Depression					
First administration	34	0.83	0.81	0.65	0.90
Second administration	34				
DASS-21 Anxiety					
First administration	34	0.83	0.81	0.66	0.90
Second administration	34				
DASS-21 Stress					
First administration	34	0.74	0.74	0.55	0.86
Second administration	34				

**Table 8 jcm-14-02097-t008:** Results of reliability analysis for the DASS-42 and DASS-21.

DASS42/DASS-21	*M*	*SD*	DASS-42	DASS-42	DASS-42	DASS-21	DASS-21	DASS-21
			Dep. *r*	Anx. *r*	Str. *r*	Dep. *r*	Anx. *r*	Str. *r*
Depression								
DASS3/DASS3	0.80	0.86	**0.70**	0.46	0.61	**0.70**	0.64	0.61
DASS5	0.75	0.83	**0.57**	0.48	0.54			
DASS10/DASS10	0.69	0.90	**0.80**	0.73	0.68	**0.78**	0.71	0.69
DASS13	0.93	0.82	0.69	0.66	**0.73**			
DASS16	0.64	0.81	**0.76**	0.70	0.72			
DASS17/DASS17	0.51	0.85	**0.76**	0.73	0.65	**0.75**	0.73	0.70
DASS21	0.55	0.79	**0.75**	0.66	0.65			
DASS24	0.72	0.81	**0.67**	0.60	0.60			
DASS26/DASS13	0.61	0.83	**0.83**	0.80	0.73	**0.80**	0.78	0.72
DASS31/DASS16	0.72	0.84	**0.77**	0.71	0.63	**0.75**	0.68	0.64
DASS34	0.40	0.71	**0.78**	0.71	0.65			
DASS37	0.54	0.77	**0.75**	0.68	0.66			
DASS38/DASS21	0.53	0.86	**0.82**	0.73	0.69	**0.81**	0.71	0.71
DASS42/DASS5	0.68	0.85	0.70	**0.72**	0.65	0.69	**0.69**	0.66
Mean inter-item *r*			0.58			0.63		
α			0.95			0.92		
Anxiety								
DASS2/DASS2	0.95	0.81	0.43	**0.46**	0.40	**0.42**	0.42	0.39
DASS4/DASS4	0.73	0.86	0.61	**0.69**	0.52	0.63	**0.64**	0.55
DASS7	0.91	0.85	0.57	**0.57**	0.55			
DASS9	1.00	0.83	0.41	0.43	**0.46**			
DASS15	0.41	0.70	0.66	**0.68**	0.53			
DASS19	0.96	0.92	0.51	**0.56**	0.51			
DASS20/DASS20	0.57	0.83	0.72	**0.73**	0.65	0.74	**0.74**	0.67
DASS23	0.48	0.74	0.63	**0.65**	0.53			
DASS25/DASS19	0.78	0.87	0.62	**0.69**	0.59	0.63	**0.66**	0.59
DASS28/DASS15	0.73	0.93	0.68	0.63	**0.71**	0.66	0.60	**0.68**
DASS30	0.60	0.76	**0.74**	0.69	0.66			
DASS36	0.37	0.67	**0.77**	0.72	0.67			
DASS40/DASS9	0.49	0.83	0.71	**0.71**	0.64	**0.74**	0.73	0.69
DASS41/DASS7	0.67	0.87	0.62	**0.68**	0.50	0.62	**0.63**	0.53
Mean inter-item *r*				0.44			0.47	
α				0.92			0.86	
Stress								
DASS1	1.09	0.76	0.53	0.50	**0.53**			
DASS6/DASS6	0.97	0.82	0.58	0.58	**0.70**	0.54	0.54	**0.65**
DASS8/DASS12	0.81	0.89	**0.60**	0.59	0.57	0.58	0.56	**0.62**
DASS11	0.92	0.87	0.66	0.63	**0.73**			
DASS12/DASS8	0.99	0.88	0.60	0.59	**0.74**	0.55	0.54	**0.65**
DASS14	0.98	0.94	0.46	0.48	**0.59**			
DASS18/DASS18	0.92	0.84	0.59	0.55	**0.67**	0.56	0.51	**0.62**
DASS22/DASS1	0.75	0.86	**0.79**	0.75	0.69	**0.78**	0.71	0.69
DASS27	10.02	0.89	0.61	0.60	**0.75**			
DASS29	0.92	0.84	0.63	0.59	**0.72**			
DASS32	0.93	0.93	0.54	0.52	**0.67**			
DASS33	0.98	0.87	0.65	0.59	**0.76**			
DASS35/DASS14	0.84	1.00	0.52	0.52	**0.56**	0.50	0.51	**0.55**
DASS39/DASS11	0.43	0.79	0.71	**0.72**	0.62	0.74	**0.75**	0.64
Mean inter-item *r*					0.48			0.47
α					0.93			0.86
Highest possible total score (%)			0.7	0.0	0.0	4.3	0.0	0.0
Lowest possible total score (%)			11.4	3.3	3.3	18.2	8.3	9.0

*Note:* Bold items represent DASS-21 items.

## Data Availability

Data underlying this article are available from the Open Science Framework (osf.io/r4e8n).

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
