# Peer review of "Psychometric Properties of the Depression Anxiety Stress Scales (DASS-42 and DASS-21) in Patients with Hematologic Malignancies"

_jcm, 2025, doi:10.3390/jcm14062097_

Round 1
Reviewer 1 Report
Comments and Suggestions for Authors
The manuscript entitled “Psychometric Properties of The Depression, Anxiety, and Stress 2 Scales (DASS-42 and DASS-21) in Patients with Hematologic 3 Malignancies” is well structured and effectively written.
The research study collected the data from total of 452 patients diagnosed with hematologic cancer.
This study is particularly relevant to healthcare professionals, as it highlights the importance of screening and monitoring psychological distress in this patient population during evaluation and intervention planning.
The present study provides significant insights, demonstrating that the DASS-42 and DASS-21 are reliable and valid psychometric instruments suitable for accessing Turkish hematologic cancer patients.
This manuscript is expected to minor revision to enhance clarity and address specific feedback:
- Given the demographic variations, particular regarding age. So, how might these differences influence the interpretation of results?
- While the DASS-42 and DASS-21 scales primarily access depression, anxiety and stress they may have limitations in capturing other mental health concerns. Could the authors discuss these aspects?
- Both DASS-42 and DASS-21 scales are susceptible to social desirability bias and subjective interpretation of questions, potentially impacting result accuracy, could this limitation further elaborated upon?
- The authors should provide more discussion on the clinical significance and therapeutic implications of the findings.
- There are several sentences can be rephrased
Line 22: Increased risk for developing to Increased risk of developing.
Line 37: Low to high negative correlation is unclear. Does it mean strong to weak correlation?
Line 325: Acceptable fit to data to Acceptable fit to the data.
Line 327: Provided a significantly better fit to data, “a” is missing.
Line 330: Decimal should be written consistently with a leading zero.
Line 336: A suitable factor structure.
Line 746: practical important implications to practically important implications.
Line 758: Unnecessary comma before implementation.
Thanks
Author Response
Comment#1. The manuscript entitled “Psychometric Properties of The Depression, Anxiety, and Stress Scales (DASS-42 and DASS-21) in Patients with Hematologic Malignancies” is well structured and effectively written. The research study collected the data from total of 452 patients diagnosed with hematologic cancer. This study is particularly relevant to healthcare professionals, as it highlights the importance of screening and monitoring psychological distress in this patient population during evaluation and intervention planning. The present study provides significant insights, demonstrating that the DASS-42 and DASS-21 are reliable and valid psychometric instruments suitable for accessing Turkish hematologic cancer patients.
Response#1. Thanks for this positive comment. There is nothing to change for this comment.
Comment#2. This manuscript is expected to minor revision to enhance clarity and address specific feedback: Given the demographic variations, particular regarding age. So, how might these differences influence the interpretation of results?
Response#2.: Thanks for this helpful comment. We added following to Limitations section.
Another limitation pertains to the demographic variations within the sample, particularly regarding age, which ranged from 18 to 91 years across the studies. This wide age range introduces potential heterogeneity that may influence the interpretation of the results. For instance, younger patients might experience distress related to disruptions in developmental milestones (e.g., education, career, or family planning), while older patients might face distress tied to comorbidities, reduced physical resilience, or existential concerns about mortality. Such age-related differences could affect how depression, anxiety, and stress are experienced and reported on the DASS-42 and DASS-21, potentially impacting the scales’ sensitivity and specificity across age groups. Although the sample size was sufficient to support the psychometric analyses, the study did not explicitly examine age as a moderating factor, limiting our under-standing of its influence on the scales’ performance. Future research should explore age-stratified analyses or measurement invariance testing across age groups to clarify these potential effects.
Comment#3. While the DASS-42 and DASS-21 scales primarily access depression, anxiety and stress they may have limitations in capturing other mental health concerns. Could the authors discuss these aspects?
Response#3. Thanks for this helpful comment. We added following to Limitations section.
While the DASS-42 and DASS-21 exhibit robust psychometric properties for assessing depression, anxiety, and stress, their scope is inherently limited in capturing the full spectrum of psychological distress experienced by hematologic cancer patients. These instruments do not specifically assess cancer-related concerns such as fear of recurrence, existential distress, body image issues, or treatment-specific anxieties, which are often prominent in this population [76]. Additionally, they do not measure other relevant mental health constructs, such as post-traumatic stress symptoms, complicated grief reactions, or adjustment disorders, which may be prevalent among cancer patients facing life-threatening illness [77,78]. Furthermore, these scales may not adequately capture culturally specific expressions of distress that could be significant in Turkish populations. Future research should consider supplementing DASS assessments with cancer-specific psychological measures to achieve a more comprehensive understanding of psychological functioning in hematologic cancer patients.
Comment#4. The authors should provide more discussion on the clinical significance and therapeutic implications of the findings.
Response#4. The bifactor model findings, which highlight the interconnected nature of depression, anxiety, and stress, suggest that clinicians should implement integrated treatment approaches rather than addressing these conditions in isolation. This interconnectedness supports the use of transdiagnostic interventions such as acceptance and commitment therapy or mindfulness-based cognitive therapy, which target underlying psychological mechanisms shared across these symptom domains. For medication management, these findings suggest psychiatrists should consider pharmacotherapies with broader effects across symptom domains rather than highly targeted medications for single conditions.
Overall, these validated instruments provide oncology teams with reliable tools to inform psychological intervention planning, evaluate treatment effectiveness, and tailor supportive care approaches to the unique needs of hematologic cancer patients throughout their treatment journey
Comment#5. There are several sentences can be rephrased.
Line 22: Increased risk for developing to Increased risk of developing
Response#5. Thanks for this helpful comment. We made changes requested by reviewer.
Old version: Consequently, they are at increased risk for developing symptoms of depression, anxiety, and stress, which can significantly undermine their quality of life.
New version: Consequently, they are at increased risk of developing symptoms of depression, anxiety, and stress, which can significantly undermine their quality of life.
Comment#6. There are several sentences can be rephrased.
Line 37: Low to high negative correlation is unclear. Does it mean strong to weak correlation.
Response#6. Thanks for this helpful comment. We made changes requested by reviewer.
Old version: Moreover, DASS-42 and DASS-21 subscales demonstrated low to high negative correlations with measures of psychological well-being (happiness, well-being, life satisfaction) and high positive correlations with measures of similar constructs (depression, anxiety, and stress),thereby supporting their convergent and discriminant validity with theoretically and empirically expected correlations with external criteria.
New Version: Moreover, DASS-42 and DASS-21 subscales demonstrated weak to strong negative correlations with measures of psychological well-being (happiness, well-being, life satisfaction) and strong positive correlations with measures of similar constructs (depression, anxiety, and stress), thereby supporting their convergent and discriminant validity with theoretically and empirically expected correlations with external criteria.
Comment#7. There are several sentences can be rephrased.
Line 325: Acceptable fit to data to Acceptable fit to the data.
Response#7. Thanks for this helpful comment. We made changes requested by reviewer.
Old version: While the one-factor and two-factor models demonstrated acceptable fit to data, the correlated three-factor and bifactor models showed excellent fit to the data.
New Version: While the one-factor and two-factor models demonstrated acceptable fit to the data, the correlated three-factor and bifactor models showed excellent fit to the data.
Comment#8. There are several sentences can be rephrased.
Line 327: Provided a significantly better fit to data, “a” is missing.
Response#8. Thanks for this helpful comment. We made changes requested by reviewer.
Old version: Results of chi-square nested difference tests revealed that both the correlated three-factor and bifactor models provided significantly better fit to data compared to the one-factor and two-factor models.
New Version: Results of chi-square nested difference tests revealed that both the correlated three-factor and bifactor models provided significantly a better fit to data compared to the one-factor and two-factor model.
Comment #9. There are several sentences can be rephrased.
Line 330: Decimal should be written consistently with a leading zero
Response#9. Thanks for this helpful comment. We made changes requested by reviewer.
Old version: The item factor loadings for Model 5 of the DASS-42 ranged from 0.69 to 0.91 for the depression latent variable, .51 to .91 for the anxiety latent variable, and 0.62 to 0.89 for the stress latent variable. For the DASS-21, item factor loadings ranged from 0.76 to 0.91 for the depression latent variable, 0.50 to 0.91 for the anxiety latent variable, and .71 to .92 for the stress latent variable.
New version: The item factor loadings for Model 5 of the DASS-42 ranged from 0.69 to 0.91 for the depression latent variable, 0.51 to 0.91 for the anxiety latent variable, and 0.62 to 0.89 for the stress latent variable. For the DASS-21, item factor loadings ranged from 0.76 to 0.91 for the depression latent variable, 0.50 to 0.91 for the anxiety latent variable, and 0.71 to 0.92 for the stress latent variable.
Comment#10. There are several sentences can be rephrased. Line 336: A suitable factor structure.
Response#10. Thanks for this helpful comment. We made changes requested by reviewer.
Old version: These findings suggest that both the correlated three-factor model and the bifactor model may be suitable factor structure for the DASS-42 and DASS-21 among hematologic cancer patients.
New version: These findings suggest that both the correlated three-factor model and the bifactor model may be a suitable factor structure for the DASS-42 and DASS-21 among hematologic cancer patients.
Comment#11. There are several sentences can be rephrased.
Line 746: practical important implications to practically important implications.
Response#11. Thanks for this helpful comment. We made changes requested by reviewer.
Comment#12. There are several sentences can be rephrased. Line 758: Unnecessary comma before implementation.
Response#12. Thanks for this helpful comment. We made changes requested by reviewer.
Old version: Furthermore, the scales demonstrate good test-retest reliability, making them suitable for use in treatment processes to monitor changes in symptoms of psychological dis-tress throughout the course of cancer treatment and evaluating the effectiveness of intervention programs.
New Version: Furthermore, the scales demonstrate good test-retest reliability, making them suitable for use in treatment processes to monitor changes in symptoms of psychological dis-tress throughout the course of cancer treatment and evaluating the effectiveness of intervention program.
Comment#13. The English is fine and does not require any improvement.
Response#13. Thanks for this positive comment. There is nothing to change for this comment

Reviewer 2 Report
Comments and Suggestions for Authors
The authors in this study have done good work by studying the psychometric properties of the depression, anxiety, and stress suggesting that the DASS-42 and DASS-21 are psychometrically robust instruments for use in hematologic patients in Turkey. However the authors need to work on the following comments to make it comprehensive.
- The percentage match (writing similarity) needs to be less than 15%.
- The authors should mention the patient collection- inclusion criteria, civil status, employment, education of the patients in the tabular format.
- The authors should subcategorize the patients considered in myeloproliferative disorders and lymphoproliferative disorders so that to get an idea of patients with AML, CLL, CML and to avoid any bias in patient collection.
- The authors should improve the English and scientific language of the manuscript.
- The authors should include the latest references in introduction and discussion.
The authors should improve the English and scientific language of the manuscript.
Author Response
Comment#1. The authors in this study have done good work by studying the psychometric properties of the depression, anxiety, and stress suggesting that the DASS-42 and DASS-21 are psychometrically robust instruments for use in hematologic patients in Turkey.
Response 1. Thanks for this positive comment. There is nothing to change for this comment.
Comment#2. However, the authors need to work on the following comments to make it comprehensive. The percentage match (writing similarity) needs to be less than 15%.
Response 2. Thanks for this helpful comment. We also checked or manuscript using Turnitin, 11% of similarity stem from journal template and content of article is completely original. Similar content stem from scale informations, it is impossible to change them scientifically. For example, DASS scale response categories show similarities (0 (Did not apply to me at all) to 3 (Applied to me very much or most of the time). It is also true for other scales.
Comment#3. The authors should mention the patient collection- inclusion criteria, civil status, employment, education of the patients in the tabular format.
Response 3. Thanks for this helpful comment. We added inclusion criteria as below to Participants section. We also added Sociodemographic Profile of Study Participants section to Study 1 and give necessary information about it. Participants were selected using a convenience sampling method based on the following inclusion criteria based on the following inclusion criteria: (a) confirmed diagnosis of a hematologic malignancy (including leukemias, lymphomas, myeloproliferative neoplasms, myelodysplastic syndromes, and plasma cell disorders such as multiple myeloma), (b) age 18 years or older, (c) currently undergoing active treatment (e.g., chemotherapy, immunotherapy, targeted therapy, or supportive care), or follow-up care for a hematologic malignancy (d) ability to provide informed consent and com-plete self-report measures, and (e) fluency in Turkish language. Participants were ex-cluded if they had: (a) severe cognitive impairment or inability to comprehend the study procedures, as determined by clinical staff, (b) acute medical emergencies or critically unstable condition, (c) significant psychiatric comorbidities that would interfere with study participation, or (d) extreme physical debilitation that would prevent completion of study measures.
Comment#4. The authors should subcategorize the patients considered in myeloproliferative disorders and lymphoproliferative disorders so that to get an idea of patients with AML, CLL, CML and to avoid any bias in patient collection.
Response 4.
Thanks for this helpful comment. We construct a table in Results section and reported all cancer types.
As seen in Table 1, there were 180 (42.7) female, and 242 (57.3%) were male he-matologic cancer patients. Their ages ranged from 18 to 91 years, with a mean of 58.40 years (SD = 14.86). A large proportion of the participants were primary school gradu-ates (n = 164; 38.9%) and married (n = 325; 77%). Of these participants, 102 (24.2%) were employed, while 320 (75.8%) were unemployed. The three most commonly diag-nosed cancer types among patients were multiple myoma (n = 114; 27%), non-hodgkin lymphoma (n = 92; 21.8%) and essential thrombocytosis (n = 30; 7.1%). Disease duration ranged from 1 to 306 months, with a mean of 45.17 months (SD = 54.17). Five patients (1.2%) did not report their age, 42 (10%) did not report their hematologic cancer diagnosis, and 42 (9.7%) did not report their disease duration.
Comment #5. The authors should improve the English and scientific language of the manuscript.
Response 5. Thanks for this helpful comment. We made a professional English editing to an American company. The proofreading certificate is in annex.
Comment #6. The authors should include the latest references in introduction and discussion.
Response 6. Thanks for helpful comments. We checked our references and updated when possible.
Comment #7. The authors should improve the English and scientific language of the manuscript.
Response 7. Thanks for this helpful comment. We made a professional English editing to an American company. The proofreading certificate is in annex.
